# SGLT2 Inhibitors: A New Therapeutical Strategy to Improve Clinical Outcomes in Patients with Chronic Kidney Diseases

**DOI:** 10.3390/ijms24108732

**Published:** 2023-05-13

**Authors:** Assunta Di Costanzo, Giovanni Esposito, Ciro Indolfi, Carmen Anna Maria Spaccarotella

**Affiliations:** 1Department of Medical and Surgical Sciences, Magna Græcia University of Catanzaro, 88100 Catanzaro, Italy; assunta.dicostanzo@studenti.unicz.it; 2Division of Cardiology, Dipartimento di Scienze Biomediche Avanzate, Università degli Studi di Napoli “Federico II”, 80134 Naples, Italy

**Keywords:** SGLT2 inhibitors, type 2 diabetes mellitus, cardiovascular disease, heart failure, chronic kidney disease

## Abstract

The purpose of this manuscript is to review the effects of sodium-glucose cotransport protein 2 inhibitors (SGLT2is) in patients with chronic kidney disease according to basic mechanisms, current recommendations, and future perspectives. Based on growing evidence from randomized, controlled trials, SGLT2is have proven their benefit on cardiac and renal adverse complications, and their indications expanded into the following five categories: glycemic control, reduction in atherosclerotic cardiovascular disease (ASCVD), heart failure, diabetic kidney disease, and nondiabetic kidney disease. Although kidney disease accelerates the progression of atherosclerosis, myocardial disease, and heart failure, so far, no specific drugs were available to protect renal function. Recently, two randomized trials, the DAPA-CKD and EMPA-Kidney, demonstrated the clinical benefit of the SGLT2is dapagliflozin and empagliflozin in improving the outcome in patients with chronic kidney disease. For the consistently positive results in cardiorenal protection, the SGLT2i represents an effective treatment to reduce the progression of kidney disease or death from cardiovascular causes in patients with and without diabetes mellitus.

## 1. Introduction

Sodium-glucose cotransport protein 2 inhibitors (SGLT2is), also called gliflozins, are responsible for major paradigm shifts in therapeutic strategies for patient care. For the consistently positive results in cardiovascular and renal protection, this class of drugs has revolutionized the cure for patients suffering from diabetes, cardiovascular diseases, and chronic kidney disease.

The history of these drugs began in 2012 when the European Medicines Agency (EMA) and the Food and Drug Administration (FDA) allowed the use of sodium-glucose cotransporter protein 2 inhibitors to reduce blood glucose in subjects with type 2 diabetes mellitus (T2DM). The three approved SGLT2is (dapagliflozin, canagliflozin, and empagliflozin) for decreasing hyperglycemia, acted by reducing renal glucose reuptake and increasing urinary glucose elimination.

While it was intended to prove cardiovascular safety to gain approval, quite unexpectedly, the cardiovascular outcome trials (CVOTs, including EMPA-REG OUTCOME, CANVAS, DECLARE-TIMI-58, VERTIS CV, and SCORED) demonstrated that SGLT2is resulted in a reduction of major adverse cardiovascular events (MACE) when compared to a placebo. Notably, in 2015 the first study to demonstrate this large protective effect on the cardiovascular system was the EMPA-REG OUTCOME trial. It showed a 14% reduction in MACE, a 34% reduction in death from any cause, and a 35% reduction in hospital admission for heart failure [1]. Subsequently, DAPA-HF and EMPEROR-Reduced, two studies in patients with a history of heart failure and reduced ejection fraction (HFrEF), evaluated another therapeutic effect of SGLT2is and revolutionized heart failure therapy. Regardless of the presence or absence of T2DM, these drugs reduced hospitalization and death in patients with ejection function < 40% [2,3]. In 2021, the EMPEROR-Preserved trial results expanded the potential target population: SGLT2 inhibitors improve the prognosis of patients with chronic heart failure, despite the ejection fraction [4].

Great impact in the clinical scene had the subsequent discovery of a nephroprotective effect provided by gliflozins. Diabetes, cardiovascular disease, and renal disease are strongly interconnected. It is known that diabetes mellitus, due to metabolic and hemodynamic alterations, causes microvascular damage. In around 40% of patients, renal vascular changes lead to chronic kidney disease (CKD) described by the presence of glomerular sclerosis or fibrosis, a stable reduction in the estimated glomerular filtration rate (eGFR) or urinary protein elimination. In addition, about half of the patients with end-stage renal disease (ESRD) have diabetes and its complications. Analysis of the renal outcomes of CVOTs showed that gliflozins slow the decline of GFR, reduce the onset of microalbuminuria, and slow the worsening of proteinuria. Renal outcomes, such as cardiovascular outcomes, are independent of the presence of diabetes mellitus, and the new EMPA-KIDNEY data show that SGLT2is are beneficial in patients with nephropathy and can be used in those with eGFR below 20 mL/min/1.73 m^2^. Therefore, SGLT2 inhibitors have further established themselves as an important tool to reduce the development of end-stage renal disease in patients with chronic kidney disease [5].

## 2. Sodium-Glucose Cotransporter 2 Inhibitors

The SGLT2 protein is a member of the Sodium Substrate Symporter gene Family (SSSF), a group of sodium-glucose symporter families codified by the SLC5A gene. The cotransporter acts as a glucose reabsorber in the renal cortex. The protein’s physiological function is transferring Na and glucose from the lumen into the cytoplasm of tubular cells; it is a secondary active transport mechanism that depends on the electrochemical Na^+^ current caused by Na^+^/K^+^-ATPase. Specifically, it is located in the luminal brush border in the first segments of the proximal tubule (S1 and S2) where the reuptake of approximately 90–97% of the filtered glucose prevents energy loss through glycosuria. The filtered glucose remaining is taken back up in the last segment of the proximal tubule by the SGLT1 protein [6]. At the opposite pole of the cell, at the level of the basolateral membrane, the reabsorbed glucose is then transferred from the intracellular space to the circulatory stream by the transporters GLUT2 (for cells with SGLT2) and GLUT1 (for cells with SGLT1).

The renal cotransporters functioning depends on the amount of glucose filtered and blood glucose levels. In healthy subjects, these transporters can recover all glucose filtered from urine. Indeed, all the glucose is filtered and subsequently reabsorbed, and the quantity of glucose eliminated in the urine is equal to 0–15 mg/dL. Specifically, the renal glucose threshold (TmaxG) refers to the value of glucose at which glucose is no longer reabsorbed and therefore first appears to be eliminated in the urine. The renal threshold is equal to a filtered glucose concentration of 300–350 mg/min, which is equivalent to a blood glucose threshold value of 180–200 mg/dL. Maximum transport refers to the maximum amount of glucose that can be reabsorbed by the transporters. When sugar concentrations in the filtrate rise excessively, some glucose escapes reabsorption because all these carriers are saturated and the direct proportionality between filtered glucose and reabsorbed glucose is then lost. This phenomenon occurs when filtered glucose exceeds the TmaxG (filtered glucose > 375 mg/min and blood glucose > 180 mg/dL). In diabetes not well controlled by medical therapy, blood glucose levels exceed the maximum transport and renal threshold causing glycosuria. Subsequently, by an adaptive mechanism, there is an increase in both SGLT2/GLUT2 transporter expression and TmaxG (>400 mg/min). This mechanism leads to increased renal tubular reabsorption of filtered glucose and exacerbated hyperglycemia.

Based on the mechanism of action of the SGLT2 protein, the idea emerged that correction of hyperglycemia could be achieved by increasing glucose excretion by inhibiting the cotransporters in the kidney. This strategy was first evaluated with the discovery of phloridzin, a natural compound derived from the apple tree and the first SGLT inhibitor that blocked kidney glucose reabsorption and increased glucose excretion. The first orally available phloridzin derivative, T-1095, was developed in 1990, but its possible use in humans was limited due to its toxicity, bad selectivity, and poor intestinal absorption. Subsequently, three highly selective gliflozins were developed and approved for the inhibition of SGLT2: dapagliflozin, empagliflozin, and canagliflozin.

Two mechanisms underlie the effects of SGLT2 inhibitors on hyperglycemia. First, they inhibit the SGLT2 cotransporter: gliflozins reduce glucose reabsorption in the proximal segments of the proximal tubule, increasing glucose excretion with urine. This causes a reduction in TmaxG and the appearance of glycosuria for glycemic values below the renal threshold. Two mechanisms underlie the effects of SGLT2 inhibitors on hyperglycemia. Firstly, they inhibit the SGLT2 cotransporter: gliflozins reduce glucose reabsorption in the proximal segments of the proximal tubule, increasing glucose excretion with urine. This causes a reduction in TmaxG and the appearance of glycosuria for glycemic values below the renal threshold. The second mechanism is the achievement of consistently lower blood glucose values, which improves glucotoxicity. This constant reduction is demonstrated by the reduction in hemoglobin A1c (HbA1c) values. In 2013, a meta-analysis of 58 studies described a favorable effect of SGLT2 inhibitors on the reduction of hemoglobinA1c levels of 0.6–1% compared to placebo and active drugs (mean difference to placebo, −0.66% [95% CI, −0.73% to −0.58%]; mean difference to active comparators, −0.06% [95% CI, −0.18% to 0.05%]) [7]. This effect improves pancreatic β-cell function and reduces insulin resistance in peripheral tissues [8].

Natriuresis is linked to the drug’s mechanism of action: the protein is a cotransporter and the inhibition of SGLT2 causes sodium elimination with urine. Urinary excretion of sodium causes the elimination of water due to an osmotic effect and consequently a decrease in blood volume. This reduction in plasma volume causes a slight dip in blood pressure of 3–6 mmHg in systolic pressure and 1–1.5 mmHg in diastolic pressure. The antihypertensive action is greater than that observed in thiazide diuretics. Initially, this drop in blood pressure and plasma volume causes a small decline in eGFR (about 5 mL/min/1.73 m^2^), which tends to return to the pre-treatment value within about 6 months [9] (Figure 1).

The role of gliflozins in diabetes therapy is undisputed, but the results of several clinical trials that have demonstrated extraordinary benefits independent of glycaemic control for patients at high cardiovascular risk and/or with renal disease are surprising.

Over 6–12 months, the additional effects of SGLT2is are improved albuminuria and proteinuria due to reduced podocyte dysfunction and alterations in the renin–angiotensin–aldosterone system (RAAS); reduced plasma uric acid levels; improved hemoglobin; weight loss of around 3 kg; and reduced cellular lipotoxicity, glucotoxicity, and oxidative stress. They demonstrated significant reductions in major cardiovascular outcomes, such as mortality and hospital admissions for heart failure, and major renal outcomes, such as progression of albuminuria and reduction of eGFR. Of particular interest is the discovery of the cardioprotective and nephroprotective role of SGLT2 independently of the presence of diabetes as a central pathology (Figure 2).

At the phenotypic level, hyperglycemia causes an alteration in endothelial homeostasis with micro- and macrovascular changes typical of the diabetic patient. At the molecular level, hyperglycemia causes an increase in the activity of protein kinase C (PKC), a protein involved in the mechanisms of oxidative stress, endothelial dysfunction, and vascular inflammation. The PKC metabolic pathway activates endothelial NO synthase (eNOS) and proinflammatory cytokines. SGLT2is are involved in improving endothelial function [10]. The main molecular mechanisms underlying this protective effect appear to be a consequence of the reduction in glucotoxicity as an effect of blood glucose reduction. It is still controversial whether the beneficial effects on the endothelium are direct or indirect.

Reduction of vascular oxidative stress. In diabetic patients, hyperglycemia causes hyperactivation of eNOS with the production of reactive oxygen species (ROS) and rapid inactivation of NO. The antioxidant effect appears to be related to the reduction of NADPH oxidase activity, resulting in improved eNOS activity.Reduction of endothelial dysfunction. Through PKC, the activity of NF-kB increases resulting in increased synthesis of inflammatory cytokines and expression of adhesion molecules on the endothelium such as IL-6, monocyte chemotactic protein-1 (MCP-1), and intercellular adhesion molecule-1 (ICAM-1). SGLT2is have been shown to reduce the secretion of these molecules in vitro and in vivo models.Reduction of the inflammatory pathway. Abundant evidence has shown that gliflozins appear to reduce the mechanisms of sclerosis and fibrosis in myocardial and renal tissue by two mechanisms. Firstly, these drugs reduce mediators of inflammation (such as IL-6, TNF, IFNγ, NF-κβ, TLR-4, and TGF-β) and growth signals for myofibroblasts in the mesangium and myocardium tissues [11]. Secondly, they improve tissue oxygenation and enhance mitochondrial function [12]. In particular, the reduction of oxidative stress results in reduced mitochondrial production of superoxide anion (O_2_^−^) and increased tissue oxygen availability.Vasodilation. This effect is a consequence of membrane hyperpolarization due to the activation of potassium channels, which causes vasodilation and reduced arterial stiffness. In addition, the increased availability of NO contributes to the modulation of vascular tone.

Another beneficial effect of SGLT2is is demonstrated in a mouse model of doxorubicin-induced cardiomyopathy. Gliflozins attenuated the cardiotoxic effects exerted by doxorubicin on left ventricular remodeling and function in terms of EF (61.5 ± 11% vs. 49.5 ± 11%), longitudinal deformation (−17.52 ± 3% vs. −13.93 ± 5%, *p* = 0.04), and circumferential deformation (−25.75 ± 6% vs. −15.91 ± 6%, *p* < 0.001), independent of glycemic control. These results could have an impact on the development of a new strategy to reduce the cardiotoxic effects of antineoplastic therapy [13].

The major clinical advantage of using SGLT2is is related to the numerous beneficial systemic effects with a very low risk of adverse events, which are generally mild. It appears from drug safety studies that the incidence of adverse events observed in patients treated with gliflozins was similar to subjects taking a placebo. The most common adverse event was genital and urinary tract mycotic and bacterial infections. The elimination of glucose with urine, especially in diabetic subjects, results in up to a fourfold increase in the frequency of infections. They are generally mild or moderate and respond to standard treatment without the need to discontinue the drug. Patients should be advised to monitor signs and symptoms and maintain proper genital hygiene [14]. Rare cases of necrotizing fasciitis of the perineum (Fournier’s gangrene), a severe complication of a urogenital infection or perineal abscess in patients with diabetes mellitus treated with SGLT2, have been reported. Alarming symptoms are fever with pain, tenderness, erythema, or swelling in the genital or perineal area. From long-term clinical trials, another rare but serious adverse event is the increased risk of amputations with a high hazard ratio for both minor (hallux and transmetatarsal) and major (ankle, above and below the knee) interventions. The results of clinical trials are controversial, and the true association is still debated in the literature. A recent review in 2022 analyzed 18 trials evaluating the association between SGLT2is and lower limb amputations. Only 2 studies on canagliflozin (CANVAS and CANVS-Renal) suggested excessive amputation rates with canagliflozin compared to placebo (6.3 vs. 3.4 per 1000 patient-years; HR 1.97; 95% CI 1.41–2.75). However, this risk was largely determined by known risk factors for amputation such as previous amputation, peripheral vascular disease (PAD: 12.1 vs. 8.2), male gender, renal disease and albuminuria, neuropathy, HbA1c > 8.0%, and the presence of cardiovascular disease. From a methodological point of view, it appeared that the remaining studies reported inconsistent and approximate results. The three main limitations were that they did not assess the prevalence and stage of PAD at baseline; the endpoints of amputation were not defined, so the severity and cause of limb events were not reported; and the CANVAS safety results and the 2017 EMA and FDA warning, resulted in selection bias with a reduced enrolment of patients with PAD [15]. Even with the controversial results, strict monitoring and routine foot care are crucial in diabetic patients. The peculiar mechanism of action related to glucose reduction by SGLT2 inhibition is not significantly associated with the risk of hypoglycemic events, and these drugs have proven to be safe with obvious clinical relevance. Furthermore, this event is encountered more frequently in patients treated with sulphonylureas or insulin. Increased osmotic diuresis causes a contraction of plasma volume, a slight reduction in blood pressure, and a transient dip in eGFR, so adequate water intake must be maintained during treatment. Hypovolaemia is a rare adverse effect but is more frequently observed in elderly patients and those treated with diuretics. In some clinical studies, a relevant but rare adverse effect described in 20 cases was diabetic ketoacidosis. These were patients with diabetes mellitus treated with insulin. Therefore, gliflozins should not be used in patients with type 1 diabetes mellitus who have a higher risk of having this serious complication.

## 3. Prevention of Cardiovascular Disease

In patients affected by diabetes mellitus, cardiovascular mortality accounts for 80% of the causes of death, but proper control of hyperglycemia can improve the prognosis of these patients, especially if started at disease onset and maintained over the long term. Given the lack of literature on the cardiovascular safety of oral hypoglycemic agents, the regulatory authorities required several large outcome studies to evaluate the benefits and security of the gliflozins. In this scenario, quite unexpectedly, SLT2is showed benefits for the secondary prevention of atherosclerotic vascular events and heart failure in patients with diabetes mellitus.

Analyses of the results of cardiovascular outcome (CVOTs) studies, including EMPA-REG OUTCOME, CANVAS, DECLARE-TIMI-58, VERTIS CV (with ertugliflozin), and SCORED (with sotagliflozin), have demonstrated the positive effects of SGLT2 inhibitors in diabetic subjects with atherosclerotic cardiovascular disease. Based on the CVOTs results, these molecules, whether or not associated with metformin, are used as first-line treatment to improve cardiovascular outcomes in these high-risk patients.

The EMPA-REG OUTCOME study compared two doses of empagliflozin versus placebo in 7020 subjects with T2DM presenting ASCVD. It was the first large study on SGLT2 inhibitors to demonstrate a reduction in composite cardiovascular outcomes and death from any cause. The primary composite endpoint of MACE was reduced by 14% in the two groups with empagliflozin. Furthermore, in the short follow-up and subsequently in the 3-year follow-up, a reduction in hospitalization for heart failure or cardiovascular death was observed in empagliflozin-treated subjects compared to placebo (HR: 0.66; 95% CI: 0.55–0.79; *p* < 0.001) [16]. The follow-up studies, CANVAS and DECLARE-TIMI 58, with canagliflozin and dapagliflozin, respectively, demonstrated similar cardiovascular benefits with these agents. CANVAS involved 10,142 participants and the primary outcome occurred in fewer patients with canagliflozin (HR 0.86; 95% CI, 0.75–0.97; *p* < 0.001) [17]. The DECLARE-TIMI 58 study evaluated 17,160 patients. In the outcome analysis, dapagliflozin was non-inferior to placebo concerning MACE, and it decreased cardiovascular death and heart failure hospitalization events (4.9% vs. 5.8%; HR 0.83; 95% CI, 0.73–0.95; *p* = 0.005) [18].

## 4. Patients with Heart Failure

Subsequently, two studies were conducted to assess the impact of these drugs on cardiovascular outcomes in patients with heart failure and reduced ejection fraction, regardless of the presence or absence of diabetes. The first study was DAPA-HF, which analyzed the effects of dapagliflozin versus placebo, in addition to optimal medical therapy (OMT), on symptom worsening or mortality in 4744 patients with chronic heart failure, ejection fraction < 40%, and elevated plasma NT-proBNP. After a follow-up of approximately 18 months, dapagliflozin therapy resulted in a 26% reduction in the primary composite endpoint [2]. Further evidence of the favorable effects of these drugs in HFrEF patients came from the EMPEROR-Reduced study: after a follow-up of about 16 months, empagliflozin in addition to standard treatments reduced the combined primary endpoint in 25% of the 3730 patients with LVEF < 40% [19]. Therefore, based on these surprising results, SGLT2is are recommended in adults for the treatment of symptomatic chronic heart failure with reduced ejection fraction.

Furthermore, recent studies have shown that gliflozins reduce the risk of cardiovascular death or hospitalization for heart failure in patients with heart failure with mildly reduced (HFmrEF) or preserved ejection fraction (HFpEF). The EMPEROR-Preserved study evaluated 5988 patients and demonstrated that empagliflozin reduced the combined risk of cardiovascular death, hospitalization for heart failure, and emergency or urgent visit for heart failure requiring intravenous treatment in patients with heart failure and ejection fraction ≥ 40% [20]. An improvement in health status and quality of life (calculated with the Kansas City Cardiomyopathy Questionnaire) has also been documented. The use of dapagliflozin (DELIVER study) also showed a 22% reduction in combined risk in 6263 patients with heart failure and mildly reduced or preserved ejection fraction [21]. Further data on its effect on inpatient and outpatient heart failure events are needed.

The EMPULSE study revealed the clinical benefits of early in-hospital initiation with empagliflozin in 530 subjects (265 in each arm) with acute heart failure regardless of left ventricular ejection fraction. In this study, empagliflozin compared to placebo was associated with a significant clinical benefit at 90 days [22].

## 5. Effects on the Renal System

Type 2 diabetes mellitus, the heart, and the kidneys are inextricably linked in terms of hemodynamic and regulatory functions. Kidney disease accelerates the progression of atherosclerosis, myocardiopathy, and heart failure. CKD is defined as the presence of eGFR < 60 mL/min/1.73 m^2^ or evidence of renal damage. The presence of albumin and protein in the urine is also the most important prognostic factor for the rapid progression of CKD to end-stage renal disease (ESRD), defined by eGFR values < 15 mL/min/1.73 m^2^ or by the need for renal replacement therapy. Microalbuminuria is defined as a random urinary albumin/Cr ratio (ACR) between 30 and 300 mg/g. An ACR above 300 mg/g is considered severe proteinuria [23]. Both eGFR and the degree of albuminuria contribute independently to the risk of future kidney injury, myocardial infarction, stroke, HF, and death. The high cardiovascular risk of people on hemodialysis (HD) is also known. In particular, repeated contact of blood with biocompatible dialysis membranes can induce an increase in markers of inflammation, oxidative stress, and activation of peripheral lymphomonocytes with excessive production of pro-inflammatory cytokines, especially interleukin-6 (IL-6), with a long-term acceleration of atherosclerosis processes. Recognition of the stage of CKD influences the approach to prognosis and management of many cardiovascular problems, as well as the use of many drugs [24].

Pharmacological management of patients with diabetes, kidney, and heart disease can be investigated to minimize complications and improve the risk of major renal and cardiovascular adverse events. The possible renal benefit of SGLT2is was first deduced from the secondary analysis of CVOTs. In particular, these drugs showed promise because they may improve cardiovascular and renal outcomes in patients with or without type 2 diabetes (Table 1).

The initial worsening of eGFR was originally a confounding factor. The reduction of about 5 mL/min/1.73 m^2^ in the first month later proved to be transient. Assessment of renal function during the longer follow-up indicated that eGFR decreased more slowly than in placebo-treated patients. Secondary endpoints in the CVOTs, assessed the effect on renal function by the progression of albuminuria (indicated by the urinary albumin-to-creatinine ratio, UACR), DoSC, reduction of eGFR (to <45 or <60 mL/min/1.73 m^2^), ESKD, death from renal causes, or need for renal replacement therapy (dialysis or transplantation). The results of these studies confirmed that SGLT2 inhibitors have beneficial effects on various renal parameters and reduce a composite of worsening eGFR, ESKD, or renal death by approximately 33% [25].

The CREDENCE trial, published in 2019, was the first randomized double-blind study that assigned 4401 patients with type 2 diabetes mellitus and chronic kidney disease with albuminuria to receive canagliflozin or a placebo. The study was stopped early after just over 2 years of follow-up and canagliflozin was associated with a 30% reduction in the risk of adverse renal and cardiovascular outcomes (HR 0.70; 95% CI, 0.59 to 0.82; *p* = 0.00001). In particular, the study aimed to assess the effect of canagliflozin on the renal outcome, and the results were encouraging with a 34% reduction in the risk of end-stage renal disease, doubling of creatinine level, or death from renal causes (HR 0.66; 95% CI, 0.53–0.81; *p* < 0.001). A beneficial effect was also assessed in subjects with eGFR < 45 mL/min/1.73 m^2^ assigned to canagliflozin [26].

In 2020, the DAPA-CKD study included 4304 patients with CKD treated with dapagliflozin or a placebo. The study was stopped prematurely for efficacy reasons; indeed, the main renal outcome was significantly lower in patients treated with dapagliflozin compared to those receiving a placebo, with a 39% reduction compared to the untreated group (HR: 0.61; 95% CI: 0.51–0.72; *p* < 0.001), regardless of the presence or absence of type 2 diabetes. A sub-analysis of DAPA-CKD in 293 subjects with stage 4 CKD and albuminuria, demonstrated the safety and efficacy of dapagliflozin even at lower eGFR levels; in fact, patients receiving dapagliflozin experienced a 27% reduction in the primary composite endpoint and 29% reductions in renal endpoints, 17% reductions in cardiovascular events, and 32% reductions in mortality risk, compared to placebo [27]. Although experience for renal benefit remains limited for patients with eGFR < 20 mL/min per 1.73 m^2^, it should be noted that SGLT2 inhibitors can be continued even if patients are on dialysis.

The EMPA-KIDNEY study enrolled 6609 adults, with or without diabetes, with chronic kidney disease with eGFR > 20 mL/min/1.73 m^2^ and UACR ≥ 200 mg/g, to evaluate the effects of empagliflozin treatment on renal disease progression or death from cardiovascular causes, similar to the DAPA-CKD study. The EMPA-KIDNEY study, which stopped early in March 2022, suggested that CKD patients without albuminuria also benefit from SGLT2 inhibitors and will soon significantly expand the population eligible for therapy. In patients treated with empagliflozin, the risk of renal disease progression or death from cardiovascular causes was 28% lower than with a placebo, with no particular safety concerns. In addition, the number of hospital admissions for any cause was lower in the empagliflozin group than in the placebo group (HR 0.86; 95% CI, 0.78–0.95; *p* = 0.003) [28].

The field of application of these drugs in individuals with advanced chronic kidney disease, who are particularly vulnerable to cardiovascular events and other complications, is controversial. Even in this category of patients, SLGT2is have shown benefits with no evidence of increased serious adverse events (SAEs) and non-serious adverse events (AEs). In subjects with lower renal function and treated with diuretics (with increased diuresis and plasma volume depletion), the reduction in eGFR is more pronounced. However, this evidence did not reduce the benefit on renal or cardiovascular outcomes. CREDENCE, DAPA-CKD, and EMPA-KIDNEY represent a strong victory for the nephropathy field, as they demonstrated an improvement in renal hard endpoints in patients with CKD, regardless of diabetes status. Therefore, the efficacy and safety results obtained offer physicians the possibility to extend the uses of SGLT2is in clinical practice to different categories of high-risk patients. The problem of subjects with CKD stage 5 at baseline, who were never enrolled in the DAPA-CKD or EMPA-KIDNEY studies, remains unresolved. However, we can point out that throughout the duration of the clinical study, neither dapagliflozin, empagliflozin, nor placebo was discontinued when eGFR fell < 15 mL/min/1.73 m^2^.

Real-world population studies confirmed the efficacy of SGLT2 inhibitors in routine clinical practice. CVD-REAL 3, an observational study conducted in 35,561 patients, confirmed the improvement of renal outcomes in those who started therapy with gliflozins compared to other hypoglycemic drugs. The results show that initiation of SGLT2 inhibitor therapy is associated with a reduced annual decline in eGFR (approximately 1.53 mL/min/1.73 m^2^) and a lower risk of major renal events [29].

## 6. Mechanisms of Nephroprotective Effect

Despite the results of the above-mentioned clinical studies showing beneficial effects on the kidneys and a reduction in the renal composite outcome in subjects treated with SGLT2is, the mechanisms of the nephroprotective effect are currently controversial. Although reducing glucose toxicity decreases the risks and severity of renal complications, the renal benefits offered by SGLT2is appear to go beyond their glycemic effect.

When an SGLT2i is started, glucose excretion leads to osmotic diuresis and natriuresis. Osmotic diuresis leads to a reduction in plasma volume and a decrease in blood pressure. From a natriuretic perspective, SGLT2 inhibition activates a reflex called tubuloglomerular feedback and increases sodium reuptake by the cells of the macula. This action creates an osmotic gradient that causes water to enter the cells and ATP to be reduced. The ATP is converted to adenosine which binds to adenosine A1 receptors on vascular smooth muscle cells causing vasoconstriction of the afferent arteriole and a reduction in intraglomerular pressure. This mechanism helps preserve glomerular viability. Adenosine-mediated effects on calcium fluxes could also reduce renin secretion from juxtaglomerular cells, which should reduce RAAS-mediated activity [30]. The activation of tubuloglomerular feedback, the release of sodium from the macula densa, the increase in afferent arteriole tone, and thus the reduction of intraglomerular pressure underlie the mechanism of reduction of hyperfiltration and albuminuria. In particular, SGLT2is have been shown to reduce albuminuria compared to placebo (33% dapagliflozin and 38% empagliflozin) [31]. Reduction of eGFR decreases filtration of toxic substances that can lead to oxidative stress and fibrosis (as albumin, hormones, and ROS) and is responsible for the long-term nephroprotective mechanism of these drugs. It also causes a reduction in nephron work, with less oxygen consumption in the renal cortex. This mechanism is responsible for the long-term nephroprotective mechanism of these drugs [32].

By reducing glucose reabsorption, SGLT2 inhibitors increase oxygen availability, reduce oxidative damage, and improve tissue viability. This oxygen availability does not explain the evidence that SGLT2 inhibitors increase erythropoietin production, which is usually associated with renal hypoxemia. By restoring proximal tubule epithelial cell metabolism and physiology, SGLT2 can reduce renal oxygen consumption and alter the production or signaling of hypoxia-inducible factors (HIFs). This could reduce HIF-1 activity and promote HIF-2α activity, promoting a decrease in pro-inflammatory and fibrotic factors and increasing erythropoietin [33]. Additionally, the observed reduction of nuclear factor-κB (NFκB), monocyte chemoattractant protein-1 (MCP-1), tumor necrosis factor receptor 1 (TNFR1), interleukin-6 (IL-6), matrix metalloproteinase 7 (MMP7), and fibronectin-1 (FN1) contributes to the reversal of molecular processes related to inflammation, extracellular matrix turnover, and fibrosis [34]. Furthermore, a meta-analysis of 62 clinical studies showed that treatment with an SGLT2i consistently reduced circulating uric acid concentrations in 34,941 patients. The probable mechanism is increased renal elimination of urate due to competition of extra glucose for the urate transporter GLUT9b. In this way, glucose is reabsorbed, and uric acid is eliminated. Lower blood levels of uric acid are associated with decreased renal damage and nephrolithiasis [35].

Inhibition of SGLT2 reduces renal glucose availability. This condition increases energy production from fatty acid metabolism and reduces lipotoxic cell damage. The suppression of the sodium–hydrogen exchanger (NHE) by SGLT2 inhibitors could explain the positive cardiac and renal role of these drugs. The NHE-1 isoform is present in myocardial cells, whereas NHE-3 is present in the kidneys. Their combined action explains the interdependence between cardiac and renal function. Indeed, NHE-3 activity is increased in patients with decompensation and results in diuretic resistance and an increase in natriuretic peptides. SGLT2 binds to the renal isoform and reduces its hyperactivity. It also blocks the activity of the cardiac isoform causing a reduction in intracellular sodium and calcium concentration with an improvement in myocardial contractile function. This effect is still unclear but could explain the cardiorenal protective action [36,37].

## 7. Summary

This manuscript provides a review of the randomized controlled trials conducted so far, with a focus on the renal benefits of these drugs. Based on numerous studies published in recent years, there is solid and consistent evidence demonstrating the efficacy and safety of SGLT2is in organ protection. Indeed, due to their consistently positive results in cardiorenal protection, SGLT2is have revolutionized the treatment of glycemic control, ASCVD, heart failure, and chronic kidney disease, with a huge impact on the clinical scene. The update of major international guidelines announced a ‘paradigm shift’ in the treatment of patients [38].

SGLT2is represent a unique therapeutic strategy to manage all three chronic diseases that have overlapping pathophysiology. In type 2 diabetes, SGLT2is are the drugs of choice together with metformin, especially in high-risk patients. In heart failure with reduced ejection fraction, SGLT2i therapy is now the first-line therapy in patients with or without diabetes. Furthermore, in the recent DELIVER and EMPEROR-Preserved studies, SGLT2is are useful in heart failure despite left ventricular ejection fractions, but their use in the absence of diabetes is not yet widespread. In renal disease and diabetes, SGLT2is and metformin are recommended as a first-line therapy in patients with an eGFR > 20–25 mL/min per 1.73 m^2^ (Table 2).

With the publication of further studies, additional approved indications for SGLT2is should continue to emerge. Areas of interest are the use of SGLT2is in type 1 diabetes and end-stage chronic kidney disease with eGFR < 15 mL/min/1.73 m^2^. The mechanism of action concerning the cardiovascular benefits of SGLT2i is still under investigation. The EMPACT-MI study is currently investigating the use of empagliflozin in patients who have had an acute myocardial infarction. Analysis of the results will test the possibility of reducing the risk of heart failure and death at follow-up by early initiation of gliflozins [39].

However, a large unmet need remains to reduce the residual risk in patients with cardiovascular or renal disease. The results from ongoing and future randomized, controlled trials will certainly expand the clinical indications for SGLT-2is in a large range of patients, enabling us to discover and unlock their value beyond the glycemic effects. Further research will help to better understand the extraordinary clinical benefits of SGLT2i drugs.

## Figures and Tables

**Figure 1 ijms-24-08732-f001:**
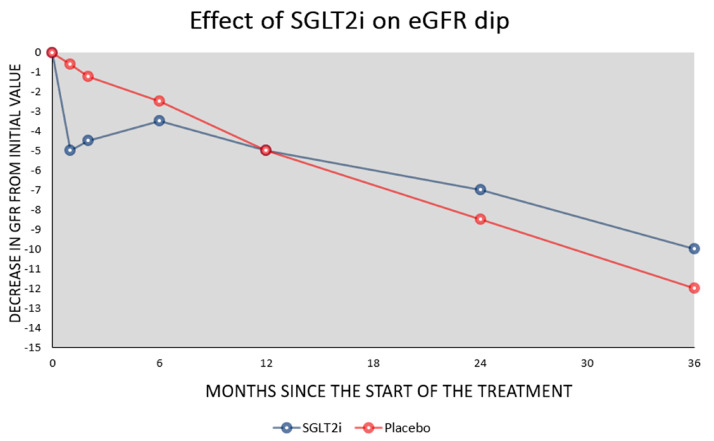
The figure is designed on data from the EMPA−REG−OUTCOME, CREDENCE, and DAPA−CKD studies. The figure shows the typical changes in eGFR caused by the use of SGLT2 inhibitors. In the first month, the eGFR dip is about 5 mL/min/1.73 m^2^ but returns towards initial values in 6 months. Thereafter the eGFR decline is slower than in subjects treated with a placebo.

**Figure 2 ijms-24-08732-f002:**
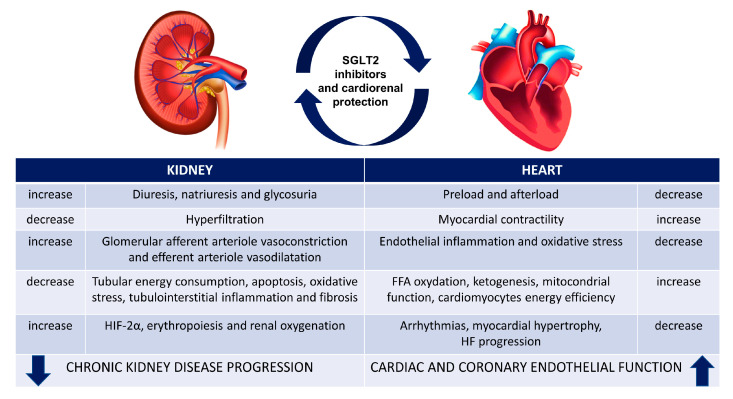
Cardiorenal protection of SGLT2i. SGLT2i have a beneficial effect: these drugs reduced chronic kidney disease progression and increased cardiac function, regardless of the presence or absence of T2DM. HIF-2α, Hypoxia-inducible factor 2α; FFA, free fatty acid; HF, heart failure.

**Table 1 ijms-24-08732-t001:** Analysis from main SGLT2i randomized controlled trials (RCT). The table presents results from the major clinical trials. Gliflozins have shown impressive benefits and demonstrated a significant reduction of main cardiovascular outcomes, such as all-cause mortality, hospitalizations for heart failure, or adverse cardiovascular events, and main renal outcomes, such as kidney disease progression, sustained decline in eGFR, or renal death.

* RCT *	Population Characteristics	Interventions	Main CV Outcomes	Main Renal Outcomes
***EMPA-REG OUTCOME*** *(NCT01131676)*	N = 7020 - Type 2 diabetes - HbA1c 7.0–9.0% without glucose-lowering therapy or HbA1c 7.0–10.0% with stable glucose-lowering therapy - Established CVD - eGFR ≥ 30 mL/min/1.73 m^2^	Empagliflozin 10 mg, Empagliflozin 25 mg, or placebo once daily	34% reduction (HR 0.83 (95% CI 0.74–0.99), *p* = 0.04) of the composite of death from cardiovascular causes, non-fatal myocardial infarction, or non-fatal stroke	39% reduction (HR 0.61 (95% CI 0.53–0.70), *p* < 0.001) of the renal-specific composite outcome *
***CANVAS*** *(NCT01032629)*	N = 10,142 - Type 2 diabetes - HbA1c 7.0–10.5% - Established CVD or two or more risk factors for CVD - eGFR ≥ 30 mL/min/1.73 m^2^	Canagliflozin 300 mg, Canagliflozin 100 mg, or placebo once daily	22% reduction (HR 0.86 (95% CI 0.75–0.97), *p* = 0.02) of the composite of death from cardiovascular causes, non-fatal myocardial infarction, or non-fatal stroke	40% reduction (HR 0.60 (95% CI 0.47–0.77), *p* < 0.001) of the renal-specific composite outcome *
***DECLARE-TIMI 58*** *(NCT01730534)*	N = 17,160 - Type 2 diabetes - HbA1c 6.5–12.0% established CVD or multiple risk factors for atherosclerotic CVD - eGFR ≥ 60 mL/min/1.73 m^2^	Dapaglifozin 10 mg, Dapaglifozin 5 mg, or placebo once daily	17% reduction (HR 0.83 (95% CI 0.73–0.95), *p* = 0.005) of the composite of death from cardiovascular causes, non-fatal myocardial infarction, or non-fatal stroke	47% reduction (HR 0.53 (95% CI 0.43–0.66), *p* < 0.0001) of the renal-specific composite outcome *
***DAPA-HF*** *(NCT03036124)*	N = 4744 - Patients with chronic HFpEF and NYHA II-IV	Dapaglifozin 10 mg, Dapaglifozin 5 mg, or placebo once daily	26% reduction (HR 0.74 (95% CI 0.65–0.85)) of the composite of cardiovascular death or HHF	26% reduction (HR 0.71 (95% CI, 0.44–1.16), *p* = 0.17) of the renal-specific composite outcome *
***EMPEROR-REDUCED*** *(NCT03057977)*	N = 3730 - Patients with chronic HFpEF and NYHA II-IV	Empagliflozin 10 mg or placebo once daily	25% reduction (HR 0.75 (95% CI, da 0.65 a 0.86), *p* < 0.001) of the composite of cardiovascular death or HHF	50% reduction (HR 0.50 (95% IC 0.32–0.77)) of the renal-specific composite outcome *
***EMPEROR-PRESERVED*** *(NCT03057951)*	N = 5988 - Patients with chronic HFpEF and NYHA II-IV - Structural heart disease or documented HHF	Empagliflozin 10 mg, or placebo once daily	21% reduction (HR 0.79, (95% CI 0.69–0.90), *p* < 0.001) of the composite of cardiovascular death or HHF	Improvement in the eGFR slope (+1.36 mL/min/1.73 m^2^ per year, 95% CI *p* < 0.001)
***CREDENCE*** *(NCT02065791)*	N = 4401 - Type 2 diabetes - ≥30 years of age - HbA1c 6.5–12.0% - Established CKD: eGFR 30–90 mL/min/1.73 m^2^ and UACR 300–5000 mg/g	Canaglifozin 100 mg or placebo once daily	31% reduction (HR 0.69 (95% CI 0.57–0.83), *p* < 0.001) of the composite of cardiovascular death or HHF	34% reduction (HR 0.66 (95% CI 0.53–0.81), *p* < 0.001) of the renal-specific composite outcome *
***DAPA-CKD*** *(NCT03036150)*	N = 4304 Inclusion Criteria: - eGFR 25–75 mL/min/1.73 m^2^ - Evidence of increased albuminuria and UACR 200–5000 mg/g	Dapaglifozin 10 mg, Dapaglifozin 5 mg, or placebo once daily	29% reduction (HR 0.71 (95% CI 0.55–0.92)) of the composite of cardiovascular death or HHF	39% reduction (HR 0.56 (95% CI 0.45–0.68)) in eGFR of at least 50%, ESKD, or renal or cardiovascular death
***EMPA KIDNEY*** *(NCT03594110)*	N = 6609 - Established CKD: eGFR 20–44 mL/min/1.73 m^2^ or 45–90 mL/min/1.73 m^2^ with UACR ≥ 200 mg/g	Empagliflozin 10 mg or placebo once daily	28% reduction in primary outcomes of kidney disease progression, sustained decline in eGFR ≥40 mg/dL, or renal or cardiovascular death (HR 0.72 (95% CI, 0.64–0.82), *p* < 0.001)	

*CVD, cardiovascular disease; HbA1c, hemoglobin A1c; eGFR, estimated glomerular filtration rate; UACR, urinary albumin-to-creatinine ratio; HFrEF, heart failure with reduced ejection function; HFpEF, heart failure with preserved ejection function; NYHA, New York Heart Association classification; HHF, hospitalization for heart failure; ESKD, end-stage kidney disease. * Renal-specific composite outcome included a composite of doubling of the serum creatinine (DoSC), ESKD, and renal death.*

**Table 2 ijms-24-08732-t002:** Indications from EMA for SGLT2is and future perspectives. Based on the inclusion criteria, drug dose, and renal function, indications for SGLT2 inhibitors have expanded into the following four categories: glycemic control, reduction in ASCVD, heart failure with reduced ejection function, and diabetic kidney disease with albuminuria. Growing evidence from randomized controlled trials proves the benefits of future use in heart failure despite the ejection function and nondiabetic CKD with albuminuria.

Indication	Criteria	Dose and Kidney Function
**Glycemic control**	- Type 2 diabetes mellitus - First-line treatment for glycemic control along with metformin	- Dapagliflozin 10 to 25 mg if eGFR ≥ 60 mL/min/1.73 m^2^ - Empagliflozin 10 mg if eGFR ≥ 60 mL/min/1.73 m^2^ or 10 mg if eGFR 30–59 mL/min/1.73 m^2^ - Canagliflozin 100 to 300 mg if eGFR ≥ 60 mL/min/1.73 m^2^ or 100 mg if eGFR 30–50 mL/min/1.73 m^2^
**Diabetic kidney disease**	- Type 2 diabetes mellitus - eGFR 25–59 mL/min/1.73 m^2^ - UACR 200–5000 mg/g	- Dapagliflozin 10 mg if eGFR ≥ 25 mL/min/1.73 m^2^ - Empagliflozin 10 mg if eGFR ≥ 30 mL/min/1.73 m^2^ - Canagliflozin 100 mg if eGFR ≥ 45 mL/min/1.73 m^2^
**Reduction in ASCVD**	- Type 2 diabetes mellitus - Established ASCVD or high risk for ASCVD	- Dapagliflozin 10 mg if eGFR ≥ 25 mL/min/1.73 m^2^ - Empagliflozin 10 mg if eGFR ≥ 30 mL/min/1.73 m^2^
**Heart failure with reduced ejection function (HFrEF)**	- NYHA classes II-IV - Elevated NT-proBNP - Reduced ejection fraction < 40%	- Dapagliflozin 10 mg if eGFR ≥ 25 mL/min/1.73 m^2^ - Empagliflozin 10 mg if eGFR ≥ 20 mL/min/1.73 m^2^
**Chronic Kidney disease**	- eGFR 25–75 mL/min/1.73 m^2^ - UACR 200–5000 mg/g	- Dapagliflozin 5 to 10 mg if eGFR ≥ 25 mL/min/1.73 m^2^
** Future Perspectives **		
**Heart failure despite the ejection fractions**	- NYHA classes II-IV - Elevated NT-proBNP - All ejection fractions	- Dapagliflozin 10 mg if eGFR ≥ 25 mL/min/1.73 m^2^ - Empagliflozin 10 mg if eGFR ≥ 20 mL/min/1.73 m^2^
**Nondiabetic kidney disease**	- Ischemic nephropathy, IgA nephropathy, FSGS, chronic pyelonephritis, and chronic interstitial nephritis - No immunosuppression in the prior 6 months - eGFR 25–75 mL/min/1.73 m^2^ - UACR 200–5000 mg/g	- Dapagliflozin 10 mg if eGFR ≥ 25 mL/min/1.73 m^2^ - Empagliflozin 10 mg if eGFR ≥ 20 mL/min/1.73 m^2^ - Canagliflozin 100 mg if eGFR ≥ 45 mL/min/1.73 m^2^

ASCVD, atherosclerotic cardiovascular disease; eGFR, estimated glomerular filtration rate; FSGS, focal segmental glomerulosclerosis; HbA1c, hemoglobin A1c; NT-proBNP, N-terminal pro-brain natriuretic peptide; NYHA, New York Heart Association; UACR, urine microalbumin-to-creatinine ratio.

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
