# Peer review of "SGLT2 Inhibitors: A New Therapeutical Strategy to Improve Clinical Outcomes in Patients with Chronic Kidney Diseases"

_ijms, 2023, doi:10.3390/ijms24108732_

Round 1

Reviewer 1 Report

Line 62 - gliflozines, not glyflozines

Line 72 - the activity of the SGLT2 protein is best described as secondary active transport

Line 87 - The concepts of renal threshold and maximal transport rate should be clearly explained to avoid confusion.

Line 90 - In diabetes mellitus, increased reabsorption of filtered glucose is a consequence, not a cause, of hyperglycemia.

Line 104 - SGLT2i do not reduce glucotoxicity by reducing HbA1c. Rather, both effects stem from the lowering of blood glucose levels.

Figure 2 - The first entry of the table is confusing: SGLTi decrease, not increase, sodium-glucose reabsorption in the proximal convoluted tubule.

Line 237 - "determined by eGFR values < 15 mL/min/1.73 m2" should be replaced by "defined by eGFR values < 15 mL/min/1.73 m2 or by the need for renal replacement therapy"

Line 273 - Please insert "need for" before "renal replacement therapy"

Line 298 - "as long as patients are on dialysis" should be replaced by, for instance, "even if patients are on dialysis".

Line 336 - The mechanisms of the nephroprotective effect of SGLT2i are currently controversial. This should be noted in a short sentence.

Moderate language editing is needed 

Reviewer 2 Report

Comments of this reviewer are:

- A comment on amputations regarding some of the SGLT-2 inhibitors should be included in the adverse events.

- Table 1: mention the type of each study (RCT)

- Table 2:  Mention if these are official indications included in the SpC and from which organizations (FDA, EMA)

Reviewer 3 Report

There are several reviews of SGLT2i and chronic kidney disease. This review is not very extensive (only 37 articles). They leave out the discussion works like those of Vallon, van Bommel, etc.

They could also add what is known about the effect of SGLT2i on the epithelium and some more molecular mechanisms to differentiate them from the existing reviews.

The bibliography of this review must be formatted (the number of authors and completed as 17.

I rescue the very informative figures and tables.

Reviewer 4 Report

I really enjoyed reading the article.

Congratulations to the authors

Author Response

Si prega di consultare l'allegato.

Round 2

Reviewer 3 Report

This work has improved substantially. As the authors say, SGLT2i is a unique therapeutic strategy for managing all three chronic diseases with overlapping pathophysiology. In addition, the benefits of using SGLT2i are superior to glycemic management, with proven cardiovascular and nephroprotective results. The findings found between SGLT2i and RAAS are very encouraging. Thus expanding the pathologies that can be treated with only one drug.

The arrangement and inclusion of references and the information collected in the tables and added texts facilitate a better understanding of the topic.